



**Anatomy of unfolding: The site-specific fold stability of Yfh1 measured by 2D NMR**
Rita Puglisi, Annalisa Pastore*, Piero Andrea Temussi*
UK-DRI at King's College London, The Wohl Institute, 5 Cutcombe Rd, SE59RT London
(UK)
*To whom correspondence should be addressed
annalisa.pastore@crick.ac.uk
temussi@unina.it
**Keywords:** biophysics, cold denaturation, NMR, protein stability, thermal unfolding,
thermodynamics
**Running title:** Site specific fold stability of Yfh1 by 2D NMR





**Abstract**
Most techniques allow detection of protein unfolding either by following the behaviour of
single reporters or as an averaged all-or-none process. We recently added 2D NMR
spectroscopy to the well-established techniques able to obtain information on the process of
unfolding using resonances of residues in the hydrophobic core of a protein. Here, we
questioned whether a detailed analysis of the individual stability curves from each resonance
could provide additional site-specific information. We used the Yfh1 protein that has the
unique feature to undergo both cold and heat denaturation at temperatures above water freezing
at low ionic strength. We show that stability curves inconsistent with the average NMR curve
from hydrophobic core residues mainly comprise exposed outliers that do nevertheless provide
precious information. By monitoring both cold and heat denaturation of individual residues we
gain knowledge on the process of cold denaturation and convincingly demonstrate that the two
unfolding processes are intrinsically different.






## Introduction

Most techniques employed to monitor protein stability are not "regiospecific", as they yield a global result, i.e. an estimate of the stability of the whole protein architecture, observable through the global evolution of secondary structure elements upon an environmental insult. This is because we postulate an all-or-none cooperative process in which the protein collapses altogether from a folded to an unfolded state. When monitoring the unfolding of a protein by CD spectroscopy, for instance, we observe intensity changes related to the disruption of α helices and/or β sheets under the influence of physical or chemical agents.

It is nevertheless interesting to gauge the response of selected regions of the protein or even single residues during the unfolding process to gain new insights into the mechanisms of unfolding in selected parts of the protein structure. A technique ideally suited for this purpose is 2D NMR spectroscopy since it permits to monitor changes in the resonances at the level of individual residues. Particularly suitable are 2D $^{15}$N HSQC spectra since they provide a direct fingerprint of the protein through mapping each of the amide protons. Volume variations of the NMR resonances may reflect changes affecting single atoms of each residue and indirectly report on how they are individually affected by the unfolding process. We have recently shown that it is possible to use 2D NMR to measure protein stability and get thermodynamics parameters comparable to those obtained by standard CD methods, provided that a suitable selection of the residues is made, followed by the subsequent average of the changes of these residues (Puglisi et al., 2020). To choose residues whose NH is deep inside the protein and relatively inaccessible to the solvent we employed SADIC (Varrazzo et al., 2005), a software that quantifies depth inside the protein (D), in combination with PopS (Cavallo et al, 2003) that yields relative accessibility at an atomic level (RA). We combined the two parameters defining a new parameter, RAD, which combines depth and exposure. We demonstrated that the stability curve calculated from averaging amide volumes from residues with a RAD value below 0.1 (here henceforth called RAD_0.1) is consistent with that calculated from CD spectroscopy (Puglisi et al., 2020).

It is now interesting to wonder what information is carried by residues far from the hydrophobic core and how they reflect the process of unfolding. This is relevant also in view of an increasing number of studies on protein stability based on the intensity variations of the resonance of a single residue upon unfolding (Danielsson et al., 2015; Smith et al., 20116; Guseman et al., 2018). The excellent agreement between NMR and CD thermodynamic parameters (Puglisi et al., 2020) put us in the position to examine the output of single residues critically, and eventually follow the process of unfolding at an atomic level.





66 Here, we present the analysis of the stability of the yeast ortholog, Yfh1, of human frataxin as

67 measured by the stability curves of most observable, isolated NH resonances. We chose Yfh1

68 to probe regiospecific unfolding because this protein is an ideal model system for measuring

69 stability curves of single residues: in addition to heat denaturation, Yfh1 has a cold denaturation

70 temperature observable above zero degrees when in the absence of salt (Pastore et al., 2007).

71 Observation of the two unfolding temperatures facilitates enormously the calculation of reliable

72 stability curves and of the whole set of thermodynamic parameters. The usefulness of Yfh1 as

73 a tool to investigate unfolding processes is evidenced not only by our subsequent work (Pastore

74 et al., 2007; Sanfelice et al., 2013; Pastore and Temussi, 2017; Martin et al., 2008; Sanfelice et

75 al., 2014; Alfano et al., 2017) but also by papers from other laboratories (Espinosa et al., 2016;

76 Chatterjee et al., 2014; Bonetti et al., 2014; Aznauryan et al., 2013).

77 We demonstrate that it is possible to sort out which individual single residues yield

78 stability curves consistent with the global unfolding process and that we can obtain valuable

79 information on the process of unfolding from residues that diverge from the average behaviour,

80 Our data also prove directly the distinct mechanisms determining the cold and heat denaturation

81 processes by providing site-specific information on solvent interactions.

82

83 **Results**

84 We collected $^{15}$N HSQC spectra of Yfh1 at different temperatures and from them plotted the

85 volumes of individual residues as a function of temperature. It is possible to extract

86 thermodynamic parameters from these plots provided some conditions are met (Privalov, 1990;

87 Martin et al., 2008). It is first assumed that unfolding transitions are two-state processes from

88 folded (F) to unfolded (U) states. It is then hypothesized that the difference of the heat capacity

89 of the two forms ($\Delta C_p$) does not depend on temperature. This assumption is considered

90 reasonable when the heat capacities of the native and denatured states change in parallel with

91 temperature variation (Privalov, 1990). When these conditions are true, the populations of the

92 two states at temperature T, $f_F(T)$ and $f_U(T)$, can be expressed as a function of the difference in

93 free energy, $\Delta G^o(T)$, according to the modified Gibbs-Helmholtz equation (Martin et al., 2008).

94 The curve corresponding to this equation is known as the stability curve of the protein (Becktel

95 and Schellman, 1987). The main thermodynamic parameters, $T_m$, $\Delta H_m$ and $\Delta C_p$, can be

96 determined using a non-linear fit (damped least-squares method). Other parameters for low

97 temperature unfolding, e.g. $T_c$, can be read from the stability curve. Volumes of isolated

98 residues were transformed into relative populations of folded Yfh1 assuming that, as found in

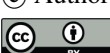



other studies on Yfh1 (Pastore et al., 2007; Martin et al., 2008; Sanfelice et al., 2014; Alfano
et al., 2017), unfolded forms are in equilibrium with a 70% population of folded Yfh1 at room
temperature (**Table 1**). The concurrent presence of an equilibrium between folded and
unfolding species is directly testified by the spectra at low ionic strength: folded and well
dispersed resonances co-exist with others strongly overlapping. These extra peaks disappear as
soon as physiologic concentrations of NaCl are added (Vilanova et al., 2014).
We then correlated each amide resonance to the corresponding value of RAD, the
parameter introduced in Puglisi et al. (2020), to pinpoint residues close to the hydrophobic
core. The behaviour of resonances in the HSQC spectrum of Yfh1 as a function of temperature
was also not uniform. While some peaks could be observed nearly at all temperatures in the
range 273-323 K, others disappeared at temperatures intermediate between room temperature
and the two unfolding temperatures, i.e. lower than 323 K or higher than 273 K (**Figure S1 of**
**Suppl. Mat.).** This behaviour was of course related to the exchange regime of these residues
and told us that they are not an integral part of the architecture of the folded form and thus their
volume variations cannot represent the all-or-none overall unfolding process faithfully. It was
anyway possible to calculate a stability curve from the temperature dependence of the
resonance volumes and the corresponding thermodynamic parameters for many residues, to
yield valuable information on the unfolding process. We then looked into what these residues
could tell us about the unfolding process.

**Residues consistent with or outliers from the global behaviour**
Comparison of the stability curves of all the well behaved residues (68 over the expected 109
resonances) with the average best curves calculated for RAD values <0.1 (henceforth called
RAD_0.1 ) showed that several residues yield stability curves drastically different from the
average curve (**Figures 1a**). The curves for residues in the hydrophobic core are overall in good
agreement with the best average curve (**Figures 1b**). However, there is in principle no clear-
cut criterion to decide when the curves are not consistent with the average. We arbitrarily chose
to set a cut-off at values of the unfolding temperatures ($T_m$ and $T_c$) that differed, on average,
less than 1.5 K from those corresponding to the average (RAD_0.1). This difference is smaller
than the variability that we had observed among different preparations and measurements of
the same protein (Pastore et al., 2007; Martin et al., 2008; Sanfelice et al., 2014; Sanfelice et
al., 2015; Alfano et al., 2017; Puglisi et al., 2020). The residues selected according to this
criterion are E71, E75, E89, L91, D101, L104, S105, M109, T110, F116, Y119, I130, L132,
A133, F142, D143, L152, L158, T159, D160, T163, and K168 (**Figure 1c**). Most of the amide





groups of the well-behaved residues are spread among well-structured secondary elements, but
there are a few in less ordered regions (**Figure 2a**). By the same token, we selected as 'ill-
behaved' residues those whose $\Delta T_m$ and $\Delta T_c$ were greater, on average, than 3 K with respect
to the best curve (RAD_0.1). Eighteen residues (V61, Q63, H83, H95, C98, G107, V108, I113,
V120, N127, K128, Q129, L136, N146, G147, N154, K172, Q174) belong to this sub-set.
Similarly, except for a few outliers, they all are in less structured regions (**Figure 2b**).

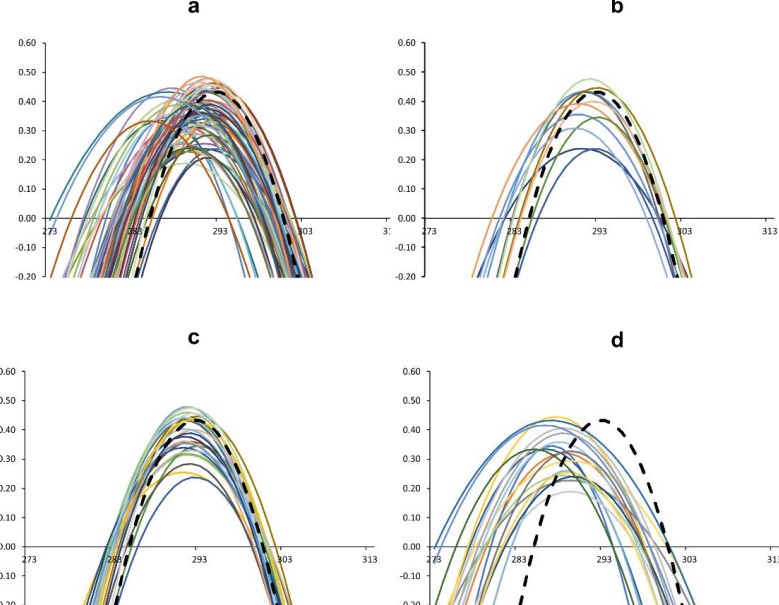


**Figure 1.** Comparison of single residue stability curves with the global (RAD_0.1) best curve
(dashed black). **a**) Stability curves of all observable isolated residues. **b**) Stability curves of
residues with a RAD<0.1. **c**) Stability curves of single residues for which the difference in the
unfolding temperatures with respect to values of the reference curve ($\Delta T_m$ and $\Delta T_c$) is on
average below 1.5 °C **d**) Stability curves of single residues for which the difference in the
unfolding temperatures with respect to values of the average curve ($\Delta Tm$ and $\Delta Tc$) is on
average above 3 K.

The stability curves of these residues (**Figure 1d**) have an important peculiarity: most stability
curves show a moderate decrease of $T_m$ and a large decrease of $T_c$. This finding is paradoxical
because it implies that the corresponding transition temperatures for the heat and cold unfolding
point to decreased and increased stability for heat and cold denaturation respectively. Although
it is difficult to explain the behaviour of the extreme values of the $\Delta T_c$ of some residues, it is
fair to say that this behaviour confirms that the mechanisms of the two unfolding processes are
intrinsically different. This possibility was already postulated by Privalov (Privalov, 1990) who



suggested that the disruption of the hydrophobic core at low temperature would be caused by
the hydration of the side chains of hydrophobic residues of the core, whereas the high
temperature transition is mainly linked to entropic reasons, consistent with the increase of
thermal motions when temperature is increased. What we observed is also in line with our
previous evidence that showed that the unfolded species at low temperature has a volume
higher than the folded species and of the high temperature unfolded species (Alfano et al.,
2017) and that cold denaturation is caused by a hydration increase (Adrover et al., 2012).

163       In light of this fundamental difference, it is tempting to hypothesize that the large $\Delta T_c$ of
residues positioned in the middle of connecting turns, G107, N127, N146 and N154 (**Figure
2c),** may reflect the fact that these flexible (well hydrated) structural elements keep some
resilience since they do not experience environmental changes even after the core has been
invaded by water molecules and are the last to be affected by unfolding.

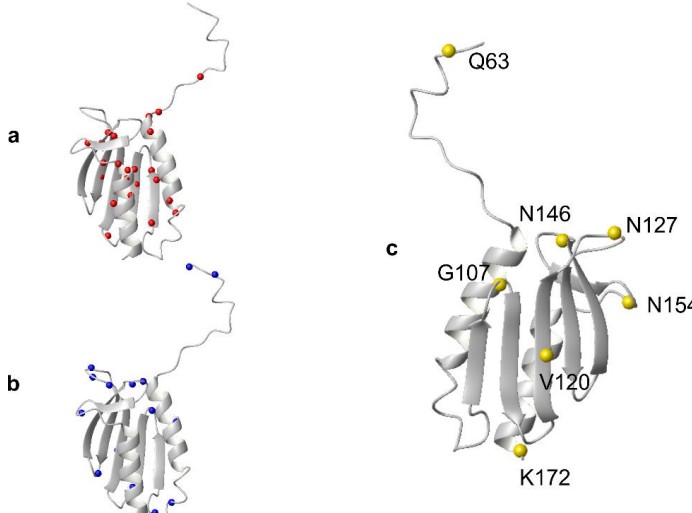


**Figure 2.** Distribution of residues on the structure of Yfh1. a) Distribution of the N atoms of
residues for which the difference in the unfolding temperatures with respect to values of the
RAD_0.1 curve ($\Delta T m$ and $\Delta T c$) is on average below 1.5 K. b) Distribution of the N atoms of
residues for which the difference in the unfolding temperatures with respect to values of the
average curve ($\Delta T m$ and $\Delta T c$) is on average above 3 K. c) The gold dots on the structure mark
the positions of residues whose stability curve is most shifted to lower temperatures with
respect to the average one (RAD_0.1).

178       In other words, these residues seem to form a kind of exoskeleton that is tougher at lower
temperatures. This view is consistent with the observation that, when decreasing the



temperature towards the cold denaturation transition there is a decrease of thermal motions that
may favour the persistence of structural elements less ordered than helices and beta sheet.

**A thermodynamic assessment of flexibility**

The negative $\Delta T_m$ and $\Delta T_c$ observed for some residues (**Figure 1d**) imply that the temperature
of maximum stability ($T_S$, so called because it corresponds to zero entropy) is lower than that
observed for the best average (RAD_0.1). The low temperature shift of $T_S$ values is consistent
with the unfolding of more flexible parts of the protein structure because it corresponds to an
increase in entropy connected to unfolding.

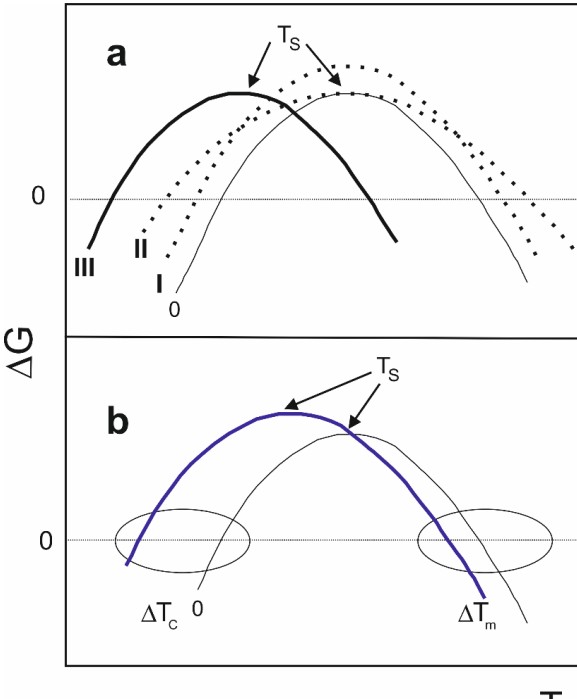


**Figure 3.** Mechanisms that influence stability curves of a protein (adapted from Nojima
et al, 1977). **a**) Dependence of the difference of free energy between unfolded and folded
states ($\Delta G$) of a hypothetical protein vs temperature (curve 0). Mechanism I illustrates
the effect of increasing $\Delta H_S$ (curve I). Mechanism II shows the effect of reducing $\Delta C_p$
(curve II). Mechanism III shows the shift of the whole stability curve towards lower
temperatures caused by increasing $\Delta S_m$ (curve III). **b**) A combination of the three
mechanisms. The solid blue curve corresponds qualitatively to the cases of Yfh1 reported
in **Figure 1 d**.

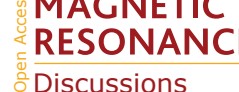

This consideration (Razvi & Scholtz, 2006) is based on the classification by Nojima et
al. (1977) of the main mechanisms of changing the thermal resistance of a protein.
Obviously, the same holds true if one wants to *decrease* $T_m$ or $T_c$: it is sufficient to reverse the
changes. According to the rough classification of Nojima et al. (1977), altered thermostability
can be achieved thermodynamically in three different ways (**Figure 3**).
According to mechanism (I), when $\Delta H_S$ (the change in enthalpy measured at $T_S$) increases, the
stability curve retains the same shape, but with greater $\Delta G$ values at all temperatures. With
mechanism II, a decreased $\Delta C_p$ leads to a broadened stability curve, because the curvature of
the stability curve is given (Becktel, & Schellman, 1987) by $\dfrac{\partial^2 \Delta G}{\partial T^2} = -\dfrac{\Delta C_p}{T}$. According to
mechanism III, the entire curve can shift towards higher or lower temperatures. It is possible
to show (Privalov, 1990) that:
$$T_S = T_m \cdot \exp\left[-\frac{\Delta S_m}{\Delta C_p}\right] = T_m \cdot \exp\left[-\frac{\Delta H_m}{T_m \cdot \Delta C_p}\right].$$

Increasing the difference in entropy between the folded and unfolded states ($\Delta S_m$) can shift
values of $T_S$ towards lower temperatures. Most of the curves of **Figure 1d** do not correspond
to a single mechanism, but to a combination of them (**Figure 3b**). However, all are shifted
toward lower values of $T_S$. The largest low-temperature differences correlate well with less
ordered regions of the structure. These regions experience largest unfolding entropies and thus
visit a larger number of conformations. It is not surprising to find this behaviour for residues
at the N- and C-termini (Q63 and K172) or in connecting loops (G107, N127, N146 and N154)
which are bound to be flexible (Halle, 2002). More surprising is, however, to find amongst
these residues also V120 which is right in the middle of the β-sheet. While we have not a
definite explanation for this observation at the moment, it could indicate a local frustration
point in this region.

**Exploring the correlation between stability and secondary structure elements**
We have previously shown that, in addition to the criteria of depth and exposition, an
alternative selection of residues over which average populations might be based on elements
of regular secondary structure (Puglisi et al., 2020). It is now possible to analyse the behaviour
of each secondary structure element. The stability curves related to accessible residues of
individual secondary structure elements are summarized in **Figures 4** and **5**. The largest



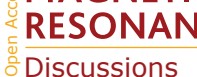

number of residues of secondary structure traits whose resonance is accessible belongs to the
two helices (**Figure 4**).

**a**

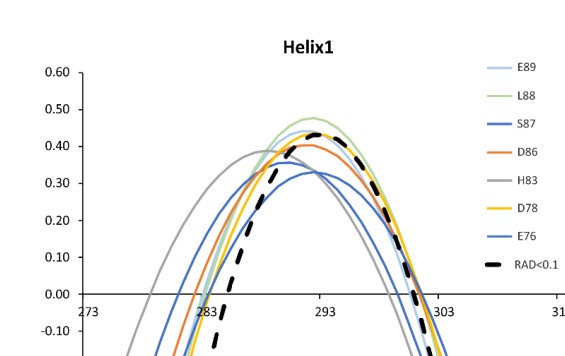

**b**

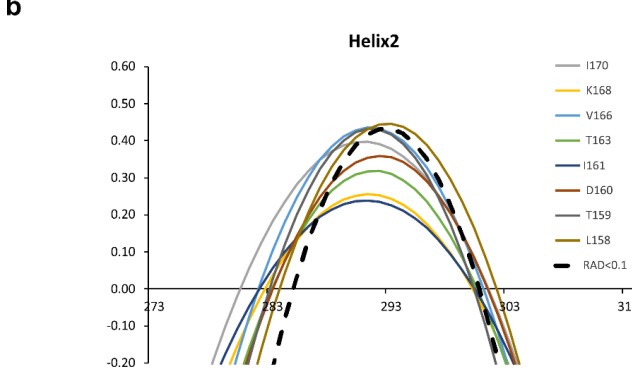


**Figure 4.** Stability curves of residues belonging to secondary structure elements. **a**) Helix 1.
**b**) Helix 2. Residues are labelled with single letter code. The average stability curve is shown
as black dashed line.

Both for helix 1 (**a**) and helix 2 (**b**) there are several resonances whose stability curve is far
from the reference one (dashed black curve of RAD_0.1). In particular, these are those of His83
and Ser87 for helix 1. All the others are in fair agreement with the average curve. The best
behaved residues (Glu76, Asp78, Leu88 and Glu89) are located at the two ends of the helix
with their amide groups in the buried side of the helix. For helix 2, the worst agreement is found
for Ile161 and Ile170, whereas the best agreement is for Leu158, Thr159, Asp160, Thr163 and
Lys168. This implies that residues of helix 2 with a good agreement are distributed over the
whole secondary structure element.

244        The number of residues belonging to beta strands for which it was possible to extract

stability curves is more limited (**Figure 5**).

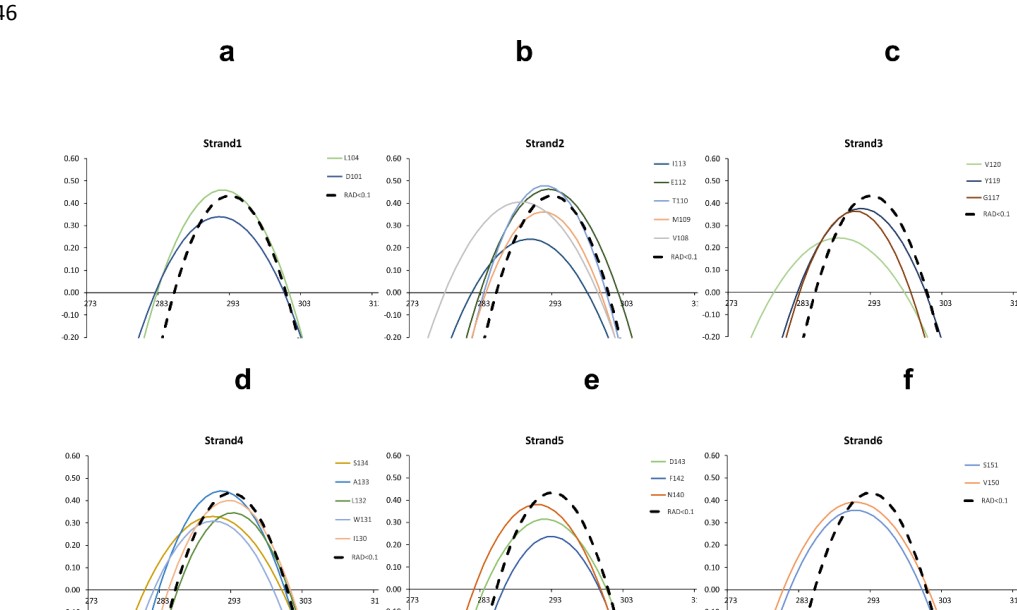

**Figure 5.** Stability curves of residues belonging to secondary structure elements. a) Strand 1.
b) Strand 2. c) Strand 3. d) Strand 4. e) Strand 5. f) Strand 6. Residues are labelled with single
letter code. The average stability curve is shown as black dashed line.

The best agreement was found for Leu104 of strand 1, Met109 and Thr110 of strand 2, Ile130,
Leu132 and Ala133 of strand 4 and Phe142 and Asp143 of strand 5. It is interesting to note
that some of the best residues reported in **Figure 1c** are not present in regular secondary
structure elements.

**The behaviour of tryptophan side chains**

We then looked into the possibility of following the process of unfolding and calculating
thermodynamic parameters using the tryptophan side chains. This choice directly parallels
studies based on following the process of unfolding by fluorescence using the intrinsic
tryptophan fluorescence (Monsellier & Bedouelle, 2002). Yfh1 has two tryptophans: W131 is
fully exposed to the solvent whereas W149 is buried. Both residues are fully conserved
throughout the frataxin family and the two side chain resonances are clearly identifiable
(**Figure S2a of Suppl. Mat.**). We calculated the thermodynamic parameters for the side chain
indole groups of both residues (**Table 1**) by the same procedure outlined for main chain NHs,
generating first a stability curve (**Figure S2b of Suppl. Mat.**). The resonance of W149, which
could potentially be more interesting, could not be used for quantitative measurements because





the temperature dependence of its volume yields a stability curve very different from the others
(**Figure S2b of Suppl. Mat.**) which leads to impossible parameters. This might be explained
by the co-existence of folded and partially unfolded species in equilibrium with each other in
solution. As a consequence the indole of W149 resonates both at 9.25 and 127.00 ppm (folded
specie) and at ca. 10.05 and 129.20 ppm (split into three closely adjacent peaks, unfolding
intermediates) (**Figure S2a of Suppl. Mat.**). As previously proven experimentally, the
resonances of the unfolding intermediates disappear upon addition of salt (Figure 1, panel A
and B in Vilanova et al., 2014). These resonances are also at the same coordinates observed for
the tryptophan indole groups at low and high temperature where however the three species
collapse into one (**Figure S1 of Suppl. Mat.**). The complex equilibrium between different
species could thus explain the ill-behaviour of the corresponding stability curve of this residue.
The behaviour of the resonance of the exposed W131 side chain is instead fully consistent with
that of RAD_0.1 and also with the original curve calculated from 1D NMR data (Pastore et al.,
2007). On the whole, these results exemplify well the complexity of the selection choice of the
unfolding reporter and advocate in favour of a wholistic analysis of the whole set of available
data.

**Discussion**
The *de facto* demonstration that it is possible to reliably measure the thermodynamic
parameters of protein unfolding by 2D NMR spectroscopy (Puglisi et al., 2020) has opened a
new territory to study protein unfolding at atomic resolution using site-specific information.
Following protein folding/unfolding looking at specific residues rather than obtaining an
average overall picture is not a novelty. Despite some intrinsic limitations, fluorescence has,
for instance, been used for decades to probe protein unfolding following the intrinsic
tryptophan fluorescence (Monsellier & Bedouelle, 2002; Bolis et al., 2004). Another elegant,
although sadly still underexploited technique able to report local behaviour at the level of
specific residues is chemically induced dynamic nuclear polarization (CIDNP) first introduced
to the study of proteins by Robert Kaptein (Kaptein et al., 1978). This technique allows the
selective observation of exposed tryptophans, histidines and tyrosines. In protein folding, it
was, for instance, used to characterize the unfolded states of lysozyme (Broadhurst et al., 1991;
Schlörb et al., 2006) and the molten globule folding intermediate of α-lactalbumin (Improta et
al., 1995; Lyon et al., 2002). Real-time CIDNP was also used to study the refolding of
ribonuclease A (Day et al., 2009) and HPr (Canet et al., 2003). The only drawback of this
technique is that, as in fluorescence, the information is limited to specific aromatic residues.



Another important technique that reports on protein unfolding at the single residue level
is stopped-flow methods coupled with NMR measurements of hydrogen exchange (Kim and
Baldwin, 1991; Roder and Wüthrich, 1986) and by mass spectrometry (Miranker et al., 1993).
In a classic paper (Miranker et al., 1991), Dobson and co-workers described, for instance, NMR
experiments based on competition between hydrogen exchange as observed in COSY spectra
and the refolding process. The authors could conclude in this way that the two structural
domains of lysozyme followed two distinct folding pathways, which significantly differed in
the extent of compactness in the early stages of folding. Similar and complementary
conclusions could be reached by integrating NMR with mass spectrometry (Miranker et al.,
1993). While these studies retain their solid importance, the possibility of following the
resonance intensities also by HSQC spectra may provide a more flexible tool to obtain detailed
information on unfolding as it reports on the exchange regime but also, implicitly, on the
chemical environment. The use of 2D HSQC had been discouraged by the non-linear
relationship between peak intensity (or volume) and the populations with temperature as the
consequence of relaxation, imperfect pulses, and mismatch of the INEPT delay with specific
J-couplings. We have previously suggested an approach to compensate for these effects and
demonstrated that the non-linearity does not affect the spectra of Yfh1 (Puglisi et al., 2020),
even though these conclusions might be protein dependent.
Here, we reconsidered our previous work (Puglisi et al., 2020) and measured individual
stability curves for most of the residues of Yfh1. Our approach showed to be particularly
fruitful for the study of this protein that has an unusual if not unique behaviour since, as a
natural unmodified full-length protein, it undergoes cold and heat denaturation when in the
absence of salts, allowing measurement of the whole stability curve. The availability of this
model system permitted us to shed light onto several important aspects.
We observed that the behaviour of the individual stability curves is not distributed
uniformly along the sequence. Residues can be clearly divided into two groups, i.e. those
consistent with the average behaviour of an all-or-none mechanism of unfolding and those
differing, even strongly, from the best average (RAD_0.1). This finding alone proved that it is
not possible to measure stability using a single residue without a careful evaluation of the role
of the specific residue in the protein fold. This conclusion is partially mitigated by our results
on the parameters obtained for a tryptophan indole. However, in the whole, also for these side
chains it may be difficult, *a priori,* to infer which tryptophan is more reliable, thus suggesting
that unfolding studies based on fluorescent measurements using the intrinsic fluorescence of
tryptophan should be taken with a pinch of salt. Even though our findings support the





possibility of obtaining protein stability parameters using the intrinsic tryptophan fluorescence
data in our specific example, in many other cases no independent controls could be done to
evaluate the accuracy of the results. The possibility of using 2D NMR and the introduction of
the easily approachable RAD parameter may assist in this choice in future studies.

340          Analysis of individual secondary structure elements, i.e. helices and strands, showed that
there is no clear hierarchy among them, and there is no indication that any of the elements
undergoes disruption before the others, either at high or at low temperature. It will be
interesting in the future to study lysozyme to have an example in which two subdomains unfold
independently (Miranker et al., 1991). In addition to information on regular secondary structure
elements, our analysis yielded also interesting information on less ordered traits. Intrinsically
flexible elements, i.e. regions characterized by multiple conformers, can be identified
unequivocally by their thermodynamic parameters, without recurring to interpretative
mechanisms.

349          Another important point is that we observed a clear difference between parameters
corresponding to the cold and the heat denaturation processes: residues that are outliers from
the average stability curve tend to have a strong stabilization effect at low temperature and a
weak destabilising effect at high temperature. This is a strong confirmation that the
mechanisms of the two transitions are intrinsically different according to the mechanism of
cold unfolding proposed by Privalov. In this model, cold denaturation is intimately linked to
the hydration of hydrophobic residues of the core (Privalov, 1990; Adrover et al., 2012). One
outstanding consequence is that, at the temperature of global unfolding, corresponding to that
of the average of the deeply buried protein core (RAD_0.1), residues outside the hydrophobic
core and in regions classified as flexible may be more resilient against unfolding. In other
words, at low temperature, opening of the hydrophobic core and its disruption can happen
before the collapse of external and more exposed elements.

361          In conclusion, we can state that monitoring protein degradation by individual residue
stability curves, as allowed by 2D NMR spectroscopy, yields a much more informative picture
than what may be obtained by traditional methods, particularly when both cold and heat
unfolding can be observed.

**Methods**
*Sample preparation*



Yeast frataxin (Yfh1) was expressed in BL21(DE3) *E. coli* as previously described (Pastore et
al., 2007). To obtain uniformly $^{15}$N-enriched Yfh1, bacteria were grown in M9 using $^{15}$N-
ammonium sulphate as the only source of nitrogen until an OD of 0.6-0.8 was reached and
induced for 4 hours at 310 K with 0.5 mM IPTG. Purification required two precipitation steps
with ammonium sulphate and dialysis followed by anion exchange chromatography using a Q-
sepharose column with a NaCl gradient. After dialysis the protein was further purified by a
chromatography using a Phenyl Sepharose column with a decreasing gradient of ammonium
sulphate.

*NMR measurements*
2D NMR $^{15}$N-HSQC experiments were run on a 700 MHz Bruker AVANCE spectrometer.
Following the strategy previously described (Puglisi et al., 2020), $^{15}$N-labelled Yfh1 was
dissolved in 10 mM Hepes at pH 7.5 to reach 0.1 mM. Spectra were recorded in the range 5-
40 ºC with intervals of 2.5 ºC and using the Watergate water suppression sequence (Piotto et
al., 1992). For each increment 8 scans were accumulated, for a total of 240 increments. Spectra
were processed with NMRPipe and analysed with CCPNMR software. Gaussian (LB -15 and
GB 0.1) and cosine window functions were applied for the direct and indirect dimension
respectively. The data were zero-filled twice in both dimensions. Spectral assignments of Yfh1
correspond to the BMRB deposition entry 19991.

**Selection of the best set of amides**
Yfh1 contains 114 backbone amide protons. The first 23 residues are an intrinsically sequence
that contains the region for mitochondrial import and processing, leading to 91 resonances in
the globular domain. 68 have non-overlapping and isolated resonances that allow easily
detectable and reliable volume calculation. Most of the excluded overlapping resonances
corresponded to disordered regions or to a partially unfolded conformation in equilibrium with
the folded one in a slow exchange regime at room temperature (Sanfelice et al., 2014).
Volumes were calculated by summation of intensities in a set box using the CCPNMR software.
The parameter RAD was used taking the parameters from the software Pops
(http://mathbio.nimr.mrc.ac.uk/~ffranca/POPS)                  and                  SADIC
(http://www.sbl.unisi.it/prococoa/).
Residues involved in secondary structures were evaluated according to the DSSP program
(https://swift.cmbi.umcn.nl/gv/dssp/). This software is all freely available. Our analysis
resulted in 35 residues in secondary structure elements (15 in alpha helices, 20 in beta sheets),



39 residues having RAD <0.5, 37 with RAD <0.4, 33 with RAD <0.3, 24 with RAD <0.2 and
11 having RAD < 0.1 (Puglisi et al., 2020).

**AUTHOR INFORMATION**
**Corresponding authors**
annalisa.pastore@crick.ac.uk
temussi@unina.it

**Acknowledgments**
The research was supported by UK Dementia Research Institute (RE1 3556) which is funded
by the Medical Research Council, Alzheimer's Society and Alzheimer's Research UK. We also
thankfully acknowledge the Francis Crick Institute for provision of access to the MRC
Biomedical NMR Centre. The Francis Crick Institute receives its core funding from Cancer
Research UK (FC001029), the UK Medical Research Council (FC001029) and the Wellcome
Trust (FC001029). We thank Geoff Kelly and Tom Frenkiel of the MRC Biomedical NMR
Centre for helpful discussions and technical support, Neri Niccolai and Franca Fraternali for
help with their software SADIC and PopS respectively. We also acknowledge the use of the
NMR spectrometers at the Randall unit of King's College London.




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

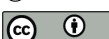

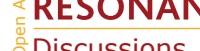

**Table 1**. Thermodynamic parameters of all residues

| | ΔH (Kcal/mol) | ΔS (Kcal/mol) | ΔCp (Kcal/molK) | Tm(K) | Tc(K) | RAD |
|---|---|---|---|---|---|---|
| 61 Val | 19.94 | 0.067 | 1.58 | 298.41 | 273.82 | 48.20 |
| 63 Gln | 21.07 | 0.072 | 2.24 | 294.03 | 275.56 | 3.64 |
| 64 Glu | 27.70 | 0.093 | 3.30 | 297.96 | 281.43 | 2.52 |
| 65 Val | 24.30 | 0.081 | 2.71 | 299.73 | 282.10 | 0.31 |
| 68 Leu | 21.01 | 0.071 | 3.02 | 296.52 | 282.77 | 0.75 |
| 70 Leu | 28.54 | 0.096 | 3.73 | 298.15 | 283.06 | 2.35 |
| 71 Glu | 29.09 | 0.097 | 3.59 | 299.79 | 283.82 | 7.13 |
| 72 Lys | 33.11 | 0.111 | 3.73 | 299.59 | 282.13 | 6.24 |
| 75 Glu | 24.30 | 0.081 | 2.75 | 300.01 | 282.63 | 2.07 |
| 76 Glu | 21.81 | 0.072 | 2.37 | 301.19 | 283.11 | 0.91 |
| 78 Asp | 29.03 | 0.096 | 3.18 | 301.13 | 283.19 | 0.17 |
| 83 His | 22.77 | 0.076 | 2.21 | 298.36 | 278.17 | 0.16 |
| 86 Asp | 25.39 | 0.084 | 2.62 | 300.96 | 281.94 | 0.27 |
| 87 Ser | 22.84 | 0.076 | 2.42 | 299.10 | 280.57 | 0.34 |
| 88 Leu | 31.31 | 0.104 | 3.38 | 301.03 | 282.83 | 0.04 |
| 89 Glu | 30.04 | 0.100 | 3.36 | 300.20 | 282.62 | 0.20 |
| 90 Glu | 26.07 | 0.087 | 2.50 | 300.86 | 280.43 | 0.52 |
| 91 Leu | 34.10 | 0.114 | 4.19 | 300.30 | 284.26 | 0.16 |
| 92 Ser | 28.77 | 0.096 | 2.93 | 300.13 | 280.87 | 0.15 |
| 93 Glu | 19.52 | 0.065 | 1.97 | 300.08 | 280.64 | 0.61 |
| 94 Ala | 23.58 | 0.079 | 2.55 | 299.58 | 281.41 | 4.10 |
| 95 His | 18.60 | 0.062 | 1.31 | 300.69 | 273.12 | 0.28 |
| 97 Asp | 23.31 | 0.078 | 2.52 | 299.02 | 280.85 | 0.95 |
| 98 Cys | 22.59 | 0.076 | 2.59 | 297.67 | 280.51 | 0.26 |
| 99 Ile | 22.05 | 0.074 | 2.58 | 298.23 | 281.41 | 0.11 |
| 101 Asp | 22.34 | 0.074 | 2.42 | 300.23 | 282.09 | 1.08 |
| 104 Leu | 29.54 | 0.098 | 3.12 | 300.93 | 282.34 | 0.78 |
| 105 Ser | 27.75 | 0.092 | 3.15 | 300.05 | 282.72 | 1.18 |
| 107 Gly | 19.28 | 0.065 | 2.44 | 296.58 | 281.00 | 3.58 |
| 108 Val | 22.33 | 0.075 | 2.03 | 299.01 | 277.49 | 0.50 |
| 109 Met | 26.64 | 0.089 | 3.25 | 299.41 | 283.26 | 0.63 |
| 110 Thr | 33.20 | 0.111 | 3.79 | 300.20 | 282.97 | 0.23 |
| 112 Glu | 28.52 | 0.094 | 2.88 | 301.86 | 282.43 | 0.41 |
| 113 Ile | 17.44 | 0.059 | 2.11 | 297.57 | 281.29 | 0.12 |
| 115 Ala | 22.32 | 0.075 | 4.01 | 297.04 | 286.00 | 2.48 |
| 116 Phe | 23.03 | 0.077 | 3.09 | 299.15 | 284.44 | 0.62 |
| 117 Gly | 27.62 | 0.093 | 3.48 | 298.20 | 282.56 | 0.98 |
| 119 Tyr | 24.94 | 0.083 | 2.72 | 300.26 | 282.24 | 0.22 |
| 120 Val | 15.70 | 0.053 | 1.68 | 297.29 | 278.94 | 0.33 |
| 127 Asn | 23.00 | 0.077 | 2.46 | 296.97 | 278.61 | 5.81 |
| 128 Lys | 15.54 | 0.052 | 1.35 | 300.36 | 277.87 | 0.66 |
| 129 Gln | 14.27 | 0.048 | 1.80 | 296.82 | 281.19 | 0.20 |
| 130 Ile | 27.87 | 0.093 | 3.19 | 300.92 | 283.73 | 0.02 |
| 131 Trp | 21.74 | 0.073 | 2.54 | 298.55 | 281.71 | 0.04 |
| 132 Leu | 26.43 | 0.088 | 3.33 | 300.68 | 285.03 | 0.02 |
| 133 Ala | 29.63 | 0.099 | 3.27 | 300.17 | 282.36 | 0.19 |
| 134 Ser | 20.31 | 0.068 | 2.07 | 299.70 | 280.45 | 0.13 |
| 136 Leu | 13.22 | 0.044 | 1.27 | 299.25 | 278.86 | 0.25 |
| 140 Asn | 25.41 | 0.085 | 2.80 | 299.43 | 281.59 | 0.17 |
| 142 Phe | 20.94 | 0.070 | 3.06 | 299.27 | 285.74 | 0.03 |
| 143 Asp | 21.86 | 0.073 | 2.50 | 300.24 | 283.04 | 0.13 |
| 146 Asn | 23.64 | 0.080 | 3.61 | 295.03 | 282.08 | 2.00 |
| 147 Gly | 25.22 | 0.085 | 2.37 | 297.81 | 276.98 | 4.80 |
| 148 Glu | 21.59 | 0.072 | 2.69 | 298.80 | 282.98 | 1.40 |
| 150 Val | 22.92 | 0.076 | 2.20 | 300.74 | 280.33 | 0.03 |
| 151 Ser | 22.70 | 0.076 | 2.39 | 299.87 | 281.22 | 0.05 |
| 152 Leu | 32.20 | 0.107 | 3.87 | 300.00 | 283.61 | 0.16 |
| 154 Asn | 21.87 | 0.074 | 2.40 | 295.07 | 277.17 | 1.14 |
| 158 Leu | 29.11 | 0.096 | 3.11 | 301.94 | 283.55 | 0.03 |
| 159 Thr | 29.76 | 0.099 | 3.38 | 300.04 | 282.72 | 0.09 |
| 160 Asp | 23.61 | 0.078 | 2.55 | 301.19 | 283.00 | 0.28 |
| 161 Ile | 15.59 | 0.052 | 1.68 | 300.08 | 281.85 | 0.09 |
| 163 Thr | 21.9 | 0.073 | 2.48 | 300.30 | 282.93 | 0.15 |
| 166 Val | 27.28 | 0.091 | 2.81 | 300.84 | 281.79 | 0.06 |
| 168 Lys | 17.28 | 0.058 | 1.93 | 299.93 | 282.33 | 0.16 |
| 170 Ile | 22.63 | 0.075 | 2.11 | 301.25 | 280.25 | 0.31 |
| 172 Lys | 28.06 | 0.095 | 3.84 | 294.06 | 279.64 | 1.5 |
| 174 Gln | 20.58 | 0.069 | 2.20 | 297.53 | 279.16 | |
| 131 Trp ε1 | 28.08 | 0.094 | 3.34 | 299.36 | 282.87 | |
