# Peer review of "Anatomy of unfolding: The site-specific fold stability of Yfh1 measured by 2D NMR"

_Magnetic Resonance, 2020_

## Referee Comment (RC1) · Anonymous Referee #1 · 24 Oct 2020

The work of Puglisi et al. deals with the observation of protein thermal denaturation processes occurring at low and high temperatures with the very interesting model of Yfh1. The authors stress the merits of 2D HSQC spectra in addressing the denaturation processes at the single-residue resolution level. This approach can surely shed light into the characteristics of the unfolding/folding transitions that may be more complex than the general all-or-none model. However the main point the authors stress, i.e. the bipartite behavior of locally structured and unstructured residues of the protein with respect to the denaturation transitions, appears really paradoxical, as the same authors point out. The intensity or volume change of the amide resonances with temperature may well indicate an unfolding transition, but may also report different processes. It may be conceivable that flexible regions of the protein could locally

anticipate the unfolding transition obtained by heating the protein, thereby providing evidence in favor of a redefinition of the all-or-none model. However it is difficult to imagine a protein exoskeleton of flexible or even locally unstructured residues that undergo the cold denaturation transition at lower temperatures with respect to the collapse of the main core. Which would be the driving forces for this "resilience", as the authors define the scenario? The authors do not provide any independent evidence supporting their interpretation. In my opinion, the lower temperature of the flexible or unstructured residue "transitions" could be interpreted as progressively slowing-down local exchange processes that eventually reach the intermediate exchange regime. These processes seem quite uncorrelated if one considers the spread of the curves in Figure 1d. The authors should at least rule out the possibility of local conformational exchange taking place in the statistically-disordered unfolded state that is achieved at Tc. The manuscript should be profoundly modified to be accepted for publication.

Please also note the supplement to this comment:
https://mr.copernicus.org/preprints/mr-2020-24/mr-2020-24-RC1-supplement.pdf

---

## Short Comment (SC1) · 24 Oct 2020

I would like to contest the conclusions drawn in mr-2020-24. In a previous article in the new commercial OA journal Commun. Chem., the authors have shown that cold denaturation of the protein Yfh1 results in the disappearance of [15N,1H]-HSQC cross-peaks of backbone amides. The main conclusions of the present article are based on the observation that the NH cross-peaks of some of the residues in less ordered structural elements seem to disappear more slowly with decreasing temperature than those of buried residues. Does this necessarily imply (as the authors suggest) that these non-core residues are less prone to cold-denaturation than residues in the core of the protein? For example, Q63 appears to be in an unstructured part of the protein. How can it unfold any further? How can a residue in the RAD_0.1 group (i.e., as I

understand, the most buried ones) be as solvent-exposed as the red ball in the top of the structure shown in Fig. 2a?

More evidence is needed to draw a conclusion that is this much at odds with common views of protein denaturation. Are there amide proton exchange data to back up the conclusions? Could the observations be explained simply as a reflection of chemical shifts that are more similar between the folded and unfolded state and, hence, less sensitive to exchange broadening? What are the exchange rates? Are the differences between residues really greater at the cold denaturation point than at the heat denaturation point? As the measurements were performed at pH 7.5, amide proton exchange with water would contribute significantly at 30 oC, i.e. cross-peaks could disappear regardless of the foldedness of the protein.

The figures and thermodynamic data of Table 1 are based on the assumption of a two-state equilibrium (folded and unfolded), but the indole resonances shown in Fig. S1 indicate the presence of more than two states. Does this not invalidate the two-state assumption?

MR strives to be a quality publication, i.e. sufficient information must be provided to validate the conclusions drawn by the authors. This implies that, even if details have been published previously, an article should be legible on its own. Which equation exactly was used to fit the amide cross-peak intensities as a function of temperature? How exactly is the fraction of folded protein derived from the cross-peak intensities? If the RAD calculations pertain to amide nitrogens, are the protein coordinates used with hydrogen atoms attached or not? Which residues exactly were determined to count as "average (RAD_0.1)"? What does RAD_0.1 actually stand for? How were the indole resonances of the unfolding intermediates assigned to W149 instead of W131? By mutation? Did the line widths of the cross-peaks vary with temperature and how did the Gaussian window used for resolution enhancement affect their intensities? Increased line widths due to chemical exchange would cause loss of 1H magnetisation during the INEPT delays of the HSQC experiment, compromising the integration of crosspeak intensities. "The behaviour of the resonance of the exposed W131 side chain is instead fully consistent with that of RAD_0.1 and also with the original curve calculated from 1D NMR data (Pastore et al., 2007)." The curves of RAD_0.1 and the original curve look quite different in Fig. S2. What is similar about them?

There are too many instances, where basic standards of care have been violated, some of which in a rather obvious manner: The main text shows figures without any units displayed on the axes. The vertical axis of the first spectrum of Fig. S1 is improperly labelled. Full references are not given in the Supplement. The names of all parameters ($T_m$, $T_c$, $\Delta H$ etc.) have not been spelled out. What is the vertical axis in Fig. 1? How is the reference curve determined? Where does the structure plotted in Fig. 2 come from? Fig. 4: which residues contribute to the average stability curve? The thermodynamic data of Table 1 are listed without explanation how they were obtained. Four significant digits suggest an accuracy that is hardly justified.

It is concerning that the primary data (i.e. peak volumes as a function of temperature) are not shown. The primary data hadn't been shown in the authors' previous article in Commun. Chem. either and there is no evidence that any new data have been recorded since.

It is suggested that special issues in MR are to fulfil the same criteria and maintain the same standard as regular articles.

---

## Referee Comment (RC2) · Dmitry Korzhnev (Referee) · 27 Oct 2020

The manuscript of Puglisi et al. describes the analysis of site-specific stability of Yfh1 based on peak integrals of the amide resonances in 1H-15N HSQC spectra. Yfh1 is a marginally stable small protein, which undergoes cold and heat denaturation at temperatures above 0C and thus represents an excellent model system to study protein folding. Building on the results of their extensive previous studies, in this paper the authors report significant differences in per-residue stability curves derived from 2D NMR for secondary structure elements and flexible loop regions. The authors offer a sound thermodynamic explanation for this behavior, which I'm tempted to believe provided they can demonstrate that variations in site-specific stability curves derived from peak integrals cannot be explained by magnetization losses during transfer periods in

1H-15N HSQC experiment due to exchange line broadening, intrinsic relaxation and exchange with the solvent.

One likely explanation of variations in temperature dependencies of peak integrals in HSQC spectra is contribution to transverse relaxation Rex due to exchange between native (N) and denatured (D) states leading to magnetization losses during transfer periods. The authors previously reported that the rate of exchange between N and D state (kex) range from 10 to 1000 1/s in the temperature range between Tc and Tm (Bonetti et al, Phys. Chem. Chem. Phys., 2014, 16, 6391-97). Considering that protein folding is accompanied by large changes in 1HN and 15N chemical shifts (dwH and dwN) and that population of the D state (pD) of Yfh1 range from 30% at the room temperature to 50% at the midpoints of transition, exchange between N and D states may result in considerable Rex even at kex of 100 1/s or slower. Due to large dwH and dwN, magnetization losses are expected to be more pronounced for NH groups from regular secondary structure elements, leading to disappearance of their signals when approaching midpoints of transition and resulting in narrower Tm-Tc range. The only rigorous way to show these effects of N-D exchange can be neglected is to estimate magnetization losses during transfer periods for each residue by numerical simulation of magnetization evolution in HSQC sequence. The authors are well equipped to do such simulations, as they already know the temperature dependences of kex and populations of N and D states, and can estimate dwH and dwN from the differences between peak positions in HSQC spectra and predicted random-coil chemical shifts.

Magnetization losses due to intrinsic 1HN and 15N spin relaxation is another factor that has to be taken into account when considering variations in temperature dependences of peak intensities in HSQC spectra. First, relaxation losses are expected to increase with decreasing the temperature due to slowing down molecular overall rotation. Second, relaxation losses are expected to be less pronounced for the flexible loop regions having slower 1HN and 15N transverse relaxation rates. Finally, amplitudes and/or time scales of internal dynamics may change with temperature and thus contribute to
temperature dependence of magnetization losses.

In addition to N-D exchange and intrinsic relaxation, HN peak intensities in the flexible protein regions at pH 7.5 may be affected by rapid amide proton exchange with water, which is expected to slow down with decreasing the temperature.

————————————————————

---

## Author Comment (AC1) · 29 Oct 2020

Reply to Prof. Otting observations First of all, we wish to thank Prof. Otting for the time spent to carefully read not only the current manuscript but also our previous production. There are however a number of possible misunderstandings which we would like to correct/clarify. 1. The possibility to interpret the data not in terms of cold denaturation. First of all we would like to clarify that there can be no doubt that Yfh1 undergoes cold as well as heat denaturation as extensively proven quantitatively by CD and 1D NMR data, that yielded identical stability curves and the first evidence of cold denaturation under physiological conditions (Pastore et al., J. Am. Chem. Soc., 129, 5374-5375 (2007)). Our work relies on 12 years of studies on cold denaturation and on at least 12 papers published in more than respectable peer reviewed journals

among which JACS (4 papers), JMB (1) and Nat. Comm. (1). Our results have been independently confirmed by other laboratories (Espinosa et al., 2016; Chatterjee et al., 2014; Bonetti et al., 2014; Aznauryan et al., 2013). Thus, our data might be interpreted in several different ways but anyway within the frame of knowing that the protein denatures at cold temperatures. 2. Two-state transitions or more. We agree with the reviewer that it is difficult to interpret the behaviour of all residues in terms of a simple two-state equilibrium, but this is not a surprise. Within the last 20 years and more, several studies have reported that the general assumption of a two-state transition breaks at the level of individual residues as for instance beautifully summarised in a recent paper by Grassein et al., 2020: "Thermal protein unfolding resembles a global (two-state) phase transition. At the local scale, protein unfolding is, however, heterogeneous and probe dependent." Both CD data and averaged NMR data can be authentically interpreted with a two-state transition, but probe dependence is precisely what we observe. The different behaviour of the side chains of W131 and W149 is a clear example: W131 is exposed and its behaviour is closer to a two-state transition. The buried W149 is instead trapped in heterogenous states. Accordingly, the curve of W149 has an impossible value of ïĄĎCp. 3. About the main conclusions of the paper and on misinterpretations. We would like to respectfully disagree with the reviewer on that "The main conclusions of the present article are based on the observation that the NH cross-peaks of some of the residues in less ordered structural elements seem to disappear more slowly with decreasing temperature than those of buried residues.". In our opinion this is not "the main conclusion". We believe that the main result is the proof that cold and heat denaturation have intrinsically different mechanisms at the single residue level. This is a very important aspect that was extensively discussed on theoretical bases by Prof. Privalov (P. Privalov, Cold denaturation of proteins. Crit ReV Biochem Mol Biol, 25: 281-305) but was not validated experimentally because of the obvious difficulties of cold denaturation studies. One of the corollaries of our results is that, surprisingly, "some of the residues in less ordered structural elements seem to disappear more slowly with decreasing temperature than those of buried

residues". This observation can indeed seem strange as we mentioned ourselves in the manuscript. We interpreted it by remembering that the main driving force of the heat denaturation mechanism is the increase of conformational entropy with temperature. This will automatically involve less ordered parts of the architecture in the unfolding process. On the contrary, cold denaturation occurs when entropy is decreasing, and the main driving force will be the sudden solvation of the hydrophobic residues of the core (P. Privalov, Cold denaturation of proteins. Crit ReV Biochem Mol Biol, 25: 281-305). As a consequence, it may happen that, while most of the (hydrophobic) core is destroyed, a few selected residues in less ordered parts keep some form of ordering. We are well aware that alternative interpretations are possible and are happy to discuss them if this can be done in a positive and constructive way (see also answer to referee 1). 4. On the possibility to use different approaches. We certainly value alternative approaches such as those alluded to by the reviewer (exchange data, chemical shift differences, exchange rates, etc. . .). We indeed discussed at length the interesting chemical shift differences observed between the cold and heat denatured states in Adrover et al., JMB 2012. However, in general these techniques pale with respect to accurate thermodynamic data calculated from the stability curves. The crucial point for studying cold denatured ensembles is the possibility to measure the full stability curve. This is possible whenever ïĄĎCp can be measured directly. Using any experimental spectroscopic technique this is not possible if cold denaturation is not accessible. It is easy to prove that fitting high temperature dependence of denaturation data is completely insensitive to the value of ïĄĎCp. When cold denaturation is accessible the value of ïĄĎCp can be measured not just predicted, on the basis of protein composition. 5. On the technical quality of our manuscript. As for the inaccuracies in the manuscript, we ensure the reviewer that we are not new to publish and that the "unacceptable" quality of the figures comes simply from the inexcusable mistake of inserting in the final manuscript the only preliminary and incomplete versions of the figures. We have now replaced them with the final ones. We also agree that papers need to be largely (although not

completely) self-explanatory but there are concepts, such as the definition of concepts such as ïĄĎH that must be common to the whole readership of this journal. We anyway sincerely thank the reviewer for bringing to our attention some unnecessary short-cuts and have done our best to improve the text. We will provide independently a detailed point-to-point list of all the changes introduced in response of the reviewer's comments. 5. On the differences of this manuscript from former work. Finally, we wish to clarify that the stress of our previous Commun. Chem. article and the present work is completely different. We wanted to explore on the compatibility of CD and NMR spectroscopies in monitoring protein unfolding and were able to show that a judicious choice of buried residues and their averaging yields almost identical thermodynamic parameters as those obtained by CD, a technique in which signal averaging is intrinsic. NMR averaging should thus not be obtained from single residues, as it can be found in the literature. After publishing the article, it occurred to us that, rather than "throwing away" the residues that misbehave from the average, we could get detailed information on why they deviate. The expert reviewer will certainly agree with us that one of the main strengths of NMR on other techniques is to provide information at the single residue level. This is the genesis of the present work. We will submit a revised version of the manuscript as soon as allowed by the editor.

Please also note the supplement to this comment:
https://mr.copernicus.org/preprints/mr-2020-24/mr-2020-24-AC1-supplement.pdf

---

## Author Comment (AC2) · 29 Oct 2020

Reply to Anonymous referee 1 (answer in bold face for clarity): The work of Puglisi et al. deals with the observation of protein thermal denaturation processes occurring at low and high temperatures with the very interesting model of Yfh1. The authors stress the merits of 2D HSQC spectra in addressing the denaturation processes at the single-residue resolution level. This approach can surely shed light into the characteristics of the unfolding/folding transitions that may be more complex than the general all-or-none model. We wish to thank the reviewer for these positive observations. Indeed, as we already mentioned in our reply to the comments of Prof. Otting, our study is not isolated and addresses a problem that has been considered for more than 30 years: whether it is possible to extract sequence-specific information

about the process of unfolding as recently spelled out by Grassein et al., J Phys Chem B 2020, 124:4391-4398: "Thermal protein unfolding resembles a global (two-state) phase transition. At the local scale, protein unfolding is, however, heterogeneous and probe dependent." However, the main point the authors stress, i.e. the bipartite behavior of locally structured and unstructured residues of the protein with respect to the denaturation transitions, appears really paradoxical, as the same authors point out. The intensity or volume change of the amide resonances with temperature may well indicate an unfolding transition, but may also report different processes. It may be conceivable that flexible regions of the protein could locally anticipate the unfolding transition obtained by heating the protein, thereby providing evidence in favor of a redefinition of the all-or-none model. However, it is difficult to imagine a protein exoskeleton of flexible or even locally unstructured residues that undergo the cold denaturation transition at lower temperatures with respect to the collapse of the main core. Which would be the driving forces for this "resilience", as the authors define the scenario? The authors do not provide any independent evidence supporting their interpretation. In my opinion, the lower temperature of the flexible or unstructured residue "transitions" could be interpreted as progressively slowing-down local exchange processes that eventually reach the intermediate exchange regime. These processes seem quite uncorrelated if one considers the spread of the curves in Figure 1d. The authors should at least rule out the possibility of local conformational exchange taking place in the statistically-disordered unfolded state that is achieved at Tc. The manuscript should be profoundly modified to be accepted for publication. We agree by and large with the referee that it is in general difficult to deal with parts of a protein with different flexibility. Our way of reasoning was the following: precisely as we cannot simply think in terms of two-state cooperative transitions when we consider thermal unfolding, high and low temperatures are not governed by the same rules. We ourselves demonstrated that the unfolded states at low temperature are different from those at high temperature (Adrover et al., Understanding cold denaturation: the case study of Yfh1. J Am Chem Soc. 132, 16240-16246. (2010); Adrover et al., The

role of hydration in protein stability: comparison of the cold and heat unfolded states of Yfh1. J. Mol. Biol. 417(5):413-24 (2012). Alfano et al., An optimized strategy to measure protein stability highlights differences between cold and hot unfolded states. Nat. Commun. 8,15428 (2017)). Here, we show that the process of unfolding itself is quite different and in full agreement with the theory published by Prof. Privalov (1990). According to this theory, the driving force of heat denaturation is the increase of conformational entropy with temperature. This will automatically disfavour less ordered parts of the architecture since they were disordered to start with. They will be those less changing. On the contrary, cold denaturation occurs when entropy is decreasing. In this case, the driving force of unfolding would be the sudden solvation of the hydrophobic residues of the core (P. Privalov, Cold denaturation of proteins. Crit ReV Biochem Mol Biol, 25: 281-305). As a consequence, it can happen that, while most of the (hydrophobic) core is destroyed, a few selected residues in less ordered parts are the last to change. In support to this hypothesis is what we observed in Adrover et al., 2010: the amide protons of the cold denatured state are ALL shifted downfield as compared to the heat denatured state. This was interpreted, as also fully supported by extensive molecular dynamics calculations, as the consequence of a more dominant effect of hydrogen bonding. Since at low temperature, hydrophobic forces are weaker, hydrogen bonds with the solvent will eventually dominate over the intramolecular hydrogen bonding. The effect that we observe in the present paper, with exposed residues undergoing cold denaturation of a lower temperature could thus reflect the fact that they are already exposed and hydrogen bonded with the solvent in the folded state. As a consequence, their volumes change less readily than resonances in the hydrophobic core that experience a more rapid all-or-none mechanism. The reviewer very helpfully suggests an alternative explanation: "the lower temperature of the flexible or unstructured residue "transitions" could be interpreted as progressively slowing-down local exchange processes that eventually reach the intermediate exchange regime." This is certainly possible, but we wonder whether we are not saying the same thing with different words. Slowing-down local exchange

processes is in fact what one would expect from a decrease of entropy and the effect of hydrogen bonding involves an exchange. Please, let us know we can agree on this point. We can easily admit that the reviewer's formulation provides a more accurate description of the phenomenon in NMR terms which could be more appropriate for the audience of this journal. We would thus be happy, if the reviewer agreed, to mention both possibilities suggesting that the two formulations might result in a different description of the same phenomenon (this is not unusual when thermodynamics concepts, that are for their very nature statistics, are described at the molecular level). Finally, the reviewer noticed that "These processes seem quite uncorrelated if one considers the spread of the curves in Figure 1d." Indeed, this is what we would expect for a process mediated by the local exchange properties of each residue.

Please also note the supplement to this comment:
https://mr.copernicus.org/preprints/mr-2020-24/mr-2020-24-AC2-supplement.pdf

---

## Author Comment (AC4) · 18 Dec 2020

**Reply to Dmitry Korzhnev**

The manuscript of Puglisi et al. describes the analysis of site-specific stability of Yfh1 based on peak integrals of the amide resonances in 1H-15N HSQC spectra. Yfh1 is a marginally stable small protein, which undergoes cold and heat denaturation at temperatures above 0C and thus represents an excellent model system to study protein folding. Building on the results of their extensive previous studies, in this paper the authors report significant differences in per-residue stability curves derived from 2D NMR for secondary structure elements and flexible loop regions.

**We are very grateful to this reviewer for his encouraging words on the thermodynamic explanation offered for per-residue stability curves.**

The authors offer a sound thermodynamic explanation for this behavior, which I'm tempted to believe provided they can demonstrate that variations in site-specific stability curves derived from peak integrals cannot be explained by magnetization losses during transfer periods in 1H15N HSQC experiment due to exchange line broadening, intrinsic relaxation and exchange with the solvent.

**We took gladly on board the helpful reviewer's comments because we realise that, even though having compensated for non-linear effects in the 2D NMR experiments, other important factors could in principle bias our data. The main challenge of the analysis is that peak intensities are affected by many factors, and these are expected to be different in the folded and in the unfolded states. Bitterly we notice that previous thermodynamic studies using HSQC, published in very high IF journals, did not address any of these concerns and no precaution was taken either for the non-linearity or for the exchange. But this is of course another matter.**

**On the plus side, however, changes in transverse relaxation during the t1 and t2 periods should not affect our conclusions, since these changes should not affect peak volumes; it is the transfer periods that matter, and perhaps the possibility that the starting magnetization might be a function of temperature if the spectra were not acquired under fully relaxed conditions (they were not).**

**The spectra in Figure S1 clearly prove that the folded and the unfolded forms are in a slow exchange regime on the NMR time scale, where peak intensities are approximately proportional to the underlying population. We have now estimated, as the reviewer suggested, the magnetisation loss during the INEPT periods during the non-enhanced $^{15}N$-$^{1}H$ HSQC experiments in collaboration with Prof. Hansen of University College London. We assumed in our simulations a two-site exchange, kex from Bonetti et al., 2015, individual populations and delta omega from the assignment of folded state and random coil. We found that the difference of populations calculated from a numerical integration of the modified Bloch-McConnell equations at low temperature is overall small. The difference is somewhat bigger at high temperature but these data are also affected by larger error because of instrumental limitations. These estimates suggest that the 'uncertainties' caused by disregarding exchange during the INEPT period is likely smaller than the spread in Fig 1c and 1d. Therefore, our conclusions about per-residue stability curves hold.**

We are certainly aware that some of these arguments should be further backed up experimentally, but a complete NMR analysis is outside the scope of the present work. Given the undeniable complexity of the multi-site system we are dealing with, our results may be the starting point for an important scientific debate which may allow the whole NMR and protein folding community to extract site-specific information about unfolding from 2D NMR spectra.